# User-Level Private Learning via Correlated Sampling

**Badih Ghazi**      **Ravi Kumar**      **Pasin Manurangsi**
Google, Mountain View, CA.
{badihghazi, ravi.k53}@gmail.com, pasin@google.com

## Abstract

Most works in learning with differential privacy (DP) have focused on the setting where each user has a single sample. In this work, we consider the setting where each user holds $m$ samples and the privacy protection is enforced at the level of each user's data. We show that, in this setting, we may learn with a much fewer number of users. Specifically, we show that, as long as each user receives sufficiently many samples, we can learn any privately learnable class via an $(\varepsilon, \delta)$-DP algorithm using only $O(\log(1/\delta)/\varepsilon)$ users. For $\varepsilon$-DP algorithms, we show that we can learn using only $O_\varepsilon(d)$ users even in the local model, where $d$ is the probabilistic representation dimension. In both cases, we show a nearly-matching lower bound on the number of users required.

A crucial component of our results is a generalization of *global stability* [BLM20] that allows the use of public randomness. Under this relaxed notion, we employ a correlated sampling strategy to show that the global stability can be boosted to be arbitrarily close to one, at a polynomial expense in the number of samples.

## 1   Introduction

Differential privacy (DP) [DMNS06, DKM$^+$06] has emerged as the accepted notion for quantifying the privacy of algorithms, whereby a method is considered private if the presence or absence of a single user has a negligible impact on its output. The two most widely-studied models of DP are the *central* model, where an analyzer has access to the raw user data, and is required to output a private answer, and the *local* model [War65, KLN$^+$11, DJW13], where the output of each user is required to be private. DP has become a widely adopted standard in both industry [EPK14, Sha14, Gre16, App17, DKY17] and government agencies, including the recent 2020 US Census [Abo18]. For a technical overview of DP, see the monographs by Dwork and Roth [DR14] and by Vadhan [Vad17].

DP has gained spotlight in machine learning (e.g., [CMS11, ACG$^+$16]), with an increased emphasis on protecting the privacy of user data used for training models. In the traditional notion of DP, the goal is to protect the privacy of each training example, where it is assumed that each user contributed precisely one such example; this is sometimes referred to as *item-level* privacy. However, a more realistic and practical setting is where a single user can contribute more than one training example. Here, the goal would be so-called *user-level* privacy, i.e., protecting the privacy of *all* the training examples contributed by a single user. This is especially relevant for federated learning settings, where each user can contribute multiple training examples [MRTZ18, WSZ$^+$19, AMR$^+$19, EMM$^+$20]; see the survey by Kairouz et al. [KMA$^+$19, Section 4.3.2], where the question of determining trade-offs between item-level and user-level DP is highlighted. It then becomes important to understand the learnability implications of this distinction between user-level vs item-level privacy.

One way to understand this problem is to artificially limit the number of training examples contributed by each user. This has been explored for some analytics and learning tasks in [AKMV19, WZL$^+$20] and is related to node-level DP [KNRS13]. While this is an interesting line of work, it does not sufficiently address the core of the problem. For instance, is learning possible with

35th Conference on Neural Information Processing Systems (NeurIPS 2021).

only a small number of users if each user contributes sufficiently many training examples? This question was addressed by Liu et al. [LSY$^+$20] for learning discrete distributions and by Levy et al. [LSA$^+$21] for some learning tasks including mean estimation, ERM with smooth losses, stochastic convex optimization. They show that the privacy cost decreases faster as the number of samples per user increases.

In this paper we address the question in a very general setting: what can user-level privacy gain for *any* privately PAC learnable class? Recall that it had recently been shown that a class is learnable via $(\varepsilon, \delta)$-DP algorithms iff it is online learnable [ALMM19, BLM20], which is in turn equivalent to the class having a finite *Littlestone dimension* [Lit87]. Furthermore, it is also known that a class is learnable via $\varepsilon$-DP algorithms iff it has a finite *probabilistic dimension* [BNS19a].

As discussed below, our protocols are based on a novel connection between *correlated sampling*—a tool from sketching and approximation algorithms [Bro97, KT02, Cha02]—and DP learning.

## 1.1 Our Results

Our first main result is that, for any online learnable class, it is possible to learn the class with an $(\varepsilon, \delta)$-DP algorithm using only $O(\log(1/\delta)/\varepsilon)$ users, as long as each user has at least $\text{poly}(d/\alpha)$ samples (Theorem 1), where $d$ is the Littlestone dimension and $\alpha$ is the error of the hypothesis output by the learner. It should be noted that the remarkable and arguably surprising aspect of this result is that we can learn using a *constant* number of samples (depending only on the privacy parameters $\varepsilon, \delta$), regardless of how complicated the class might be, as long as the class is online learnable. (For all formal definitions, see Section 2.) Indeed, previous work [AKMV19] had explicitly conjectured that such a bound is impossible for user-level learning, albeit in a different setting than ours.

**Theorem 1.** *Let $\alpha, \beta \in (0, 0.1)$, and $C$ be any concept class with finite $\text{LDim}(C) = d$. Then, for any $\varepsilon, \delta \in (0, 1)$, there exists an $(\varepsilon, \delta)$-DP $(\alpha, \beta)$-accurate learner for $C$ that requires $O\left(\log(1/(\delta\beta))/\varepsilon\right)$ users where each user has $\tilde{O}_\beta\left((d/\alpha)^{O(1)}\right)$ samples.*

Our algorithm in Theorem 1 can in fact be extended to work even in the weaker *shuffle* model of DP. We provide more detail about such an extension in Appendix D.

Our second result is a generic $\varepsilon$-DP learner in the local model for any class $C$ with finite probabilistic representation dimension $\text{PRDim}(C)$. This gives a separation in the local DP model between the user-level and item-level settings; the sample complexity in the latter is known to be polynomial in the statistical query (SQ) dimension [KLN$^+$11], which can be exponentially larger than the probabilistic representation dimension. A simple example of such a separation is PARITY on $d$ unknowns, whose probabilistic representation dimension is $O(d)$ (this can be seen by taking the class itself to be its own representation), whereas its SQ dimension is $2^{\Omega(d)}$ [BFJ$^+$94]. In this case, our Theorem 2 implies that PARITY can be learned using only $O_\beta(d/\varepsilon^2)$ users when each user has $O_{\alpha,\beta}(d^3)$ examples; by contrast, the aforementioned lower bound [KLN$^+$11] implies that, in the item-level setting (where each user has a single example), $2^{\Omega(d)}$ users are needed.

**Theorem 2.** *Let $\alpha, \beta \in (0, 0.1)$, and $C$ be any concept class with finite $\text{PRDim}(C) = d$. Then, for any $\varepsilon \in (0, 1)$, there exists an $\varepsilon$-DP $(\alpha, \beta)$-accurate learner for $C$ in the (public randomness) local model that requires $O\left(\frac{d + \log(1/\beta)}{\varepsilon^2}\right)$ users where each user has $\tilde{O}_\beta\left(d^3/\alpha^2\right)$ samples.*

In the central model, we get a slightly improved bound on the number of users in terms of $1/\varepsilon$.

**Theorem 3.** *Let $\alpha, \beta \in (0, 0.1)$, and $C$ be any concept class with finite $\text{PRDim}(C) = d$. Then, for any $\varepsilon \in (0, 1)$, there exists an $\varepsilon$-DP $(\alpha, \beta)$-accurate learner for $C$ that requires $O\left(\frac{d + \log(1/\beta)}{\varepsilon}\right)$ users where each user has $\tilde{O}_\beta\left(d^3/\alpha^2\right)$ samples.*

Interestingly, we can also show that the number of users required in the above results is essentially the smallest possible (up to a $1/\varepsilon$ factor in Theorem 2), as stated below.

**Lemma 4.** *For any $\alpha, \beta \leq 1/4, \varepsilon \in (0, 1)$ and $\delta \in (0, 0.1\varepsilon^{1.1})$, if there exists an $(\varepsilon, \delta)$-DP $(\alpha, \beta)$-accurate learner on $n$ users for a concept class $C$, then we must have $n \geq \Omega(\min\{\log(1/\delta), \text{PRDim}(C)\}/\varepsilon)$.*

The summary of our results described above can be found in Table 1.

| | Bounds | User-Level | | | Item-Level |
|---|---|---|---|---|---|
| | | # users | # samples/user | Ref. | # users |
| $\varepsilon$-DP | Upper | $O(\mathrm{PRDim}/\varepsilon)$ | $\mathrm{PRDim}^{O(1)}$ | Theorem 3 | $\Theta(\mathrm{PRDim}/\varepsilon)$ |
| (Central) | Lower | $\Omega(\mathrm{PRDim}/\varepsilon)$ | - | Lemma 4 | [BNS19a] |
| $\varepsilon$-DP | Upper | $O(\mathrm{PRDim}/\varepsilon^2)$ | $\mathrm{PRDim}^{O(1)}$ | Theorem 2 | $\mathrm{SQDim}^{\Theta(1)}$ |
| (Local) | Lower | $\Omega(\mathrm{PRDim}/\varepsilon)$ | - | Lemma 4 | [KLN$^+$11] |
| $(\varepsilon,\delta)$-DP | Upper | $O(\log(1/\delta)/\varepsilon)$ | $\mathrm{LDim}^{O(1)}$ | Theorem 1 | $\mathrm{LDim}^{O(1)}$ [GGKM21] |
| (Central) | Lower | $\Omega(\min\{\log(1/\delta),$ $\mathrm{PRDim}\}/\varepsilon)$ | - | Lemma 4 | $\Omega(\log^* \mathrm{LDim})$ [ALMM19] |

**Table 1:** Summary of our results in the user-level setting and prior results in the item-level setting. For simplicity, we assume that the accuracy parameter and success probability of the learner are constants, and we disregard their dependencies. Our lower bounds hold regardless of the number of samples each user receives.

While our previous results establish nearly tight bounds on the number of users required for DP learning, they are in general not efficient. For example, the pure-DP learners have running times that grow (at least) exponentially in the size of the probabilistic representation, and the approximate-DP learner similarly has a running time that grows (at least) exponentially in the Littlestone dimension. Our final result investigates how to get efficient learners. Informally, we show that we can take any efficient SQ algorithm and turn it into an efficient learner in the user-level setting (Theorem 5).

**Theorem 5.** *Let $C$ be any concept class, and suppose that there exists an algorithm* A *that can $\alpha$-learn $C$ using $q$ statistical queries $\mathrm{STAT}_{\mathcal{D}}(\tau)$. Furthermore, suppose that any hypothesis output by* A *can be represented by $b$ bits. Then, there exist the following algorithms:*

*(i) An $(\varepsilon,\delta)$-DP $(\alpha,\beta)$-accurate learner with $O(\log(1/\delta)/\varepsilon)$ users, where each user has* $\mathrm{poly}\left(\frac{q}{\beta\tau}\right)$ *samples.*

*(ii) An $\varepsilon$-DP $(\alpha,\beta)$-accurate learner with $O\left(\frac{b+\log(1/\beta)}{\varepsilon}\right)$ users, where each user has* $\mathrm{poly}\left(\frac{q}{\beta\tau}\right)$ *samples.*

*(iii) An $\varepsilon$-DP $(\alpha,\beta)$-accurate learner in the (public randomness) local DP model with $O\left(\frac{b+\log(1/\beta)}{\varepsilon^2}\right)$ users, where each user has* $\mathrm{poly}\left(\frac{q}{\beta\tau}\right)$ *samples.*

*Moreover, all DP learners described above run in time* $\mathrm{poly}(\mathrm{time}(\mathsf{A}), 1/\tau, 1/\beta)$.

Thanks to the abundance of SQ learning algorithms, the above result can be applied to turn those into efficient user-level DP learners. We discuss some interesting examples of these in Appendix B.2.

**Independent Work of Impagliazzo et al. [ILPS21]**

As an intermediate step of our proofs, we define a property called *pseudo-globally stability* for learning algorithms (Definition 15) and provide several such algorithms (Corollaries 21 and 27, and Lemma 32). In an independent work, Impagliazzo et al. [ILPS21] studies a similar notion under the name *reproducibility* and provide several reproducible algorithms e.g. for heavy hitters, SQ-based algorithms and learning halfspaces. Below we provide a more detailed discussion on the similarities and differences between the two papers:

- **Definition.** Strictly speaking, the main definition in [ILPS21] is slightly different compared to ours, but they note in the appendix that the two definitions are equivalent up to a polynomial factor in the parameters.
- **SQ Algorithms.** Both Impagliazzo et al.'s work and ours (Lemma 32) give generic reductions for turning SQ algorithms to pseudo-globally stable ones.
- **Amplification of Stability Parameter.** In [ILPS21, Theorem A.2], a reduction for decreasing the stability parameter is given. Indeed, one can also view our reduction in Theorem 20 in this form but our result is weaker as our reduction starts out with a (not pseudo) globally stable algorithms, whereas their reduction works even when starting with pseudo globally stable algorithms.
- **Heavy Hitter Algorithms.** Our aforementioned reduction also implicitly gives an algorithm for heavy hitters. Once again, this is weaker than that in Impagliazzo et al.: ours only gives a single heavy hitter whereas that of [ILPS21] can provide a list of all heavy hitters.

- **Additional Results in [ILPS21].** [ILPS21] also contains many additional results, including a lower bound on the overhead due to pseudo-global stability, a pseudo-globally stable median algorithm and a pseudo-globally stable algorithm for learning halfspaces with a margin.

## 1.2 Proof Overview

For simplicity of presentation, we will focus on Theorem 1; we will briefly discuss the proofs of the other results, which are similar in flavor, at the end of this section.

Let us assume for the moment that each user, given their $m$ samples drawn i.i.d. from $\mathcal{D}$, can output the same hypothesis $h^*$ (with small error) with high probability. If this holds, then we would be done: we can simply run a DP selection[1] algorithm to pick the most frequently seen hypothesis.

This assumption is quite strong, but not completely unreasonable. Specifically, Bun et al. [BLM20]—in their seminal work that characterizes hypothesis classes learnable in the item-level DP setting—showed that it is possible to come up with a learner that outputs some hypothesis $h^*$ with probability $2^{-O(d)}$, where $d$ denotes the Littlestone dimension of the concept class. We may attempt to use this in the approach described above, but this does not work: in order to even see $h^*$ at all (with say a constant probability), we would need $2^{O(d)}$ users, which is prohibitive!

To overcome this, we exploit shared randomness between users. Our main technical result here is that, if the users share randomness, we can ensure that they output the same $h^*$ with probability arbitrarily close to one; we can then run the DP selection algorithm to pick $h^*$. This immediately allows our overall strategy described above to go through.

The shared randomness is used in our algorithm(s) via *correlated sampling*. Recall that a correlated sampling strategy is an algorithm that takes in a probability distribution $\mathcal{P}$ together with randomness $r$. The guarantee is that, if we run it on two distributions $\mathcal{P}_1, \mathcal{P}_2$ but with the same randomness $r$, then the probability (over $r$) that the outputs disagree is at most a constant times the total variation distance between $\mathcal{P}_1$ and $\mathcal{P}_2$. The task then is simply to compute, for each user $i$, such a probability distribution $\mathcal{P}_i$ from their own samples such that the $\mathcal{P}_i$'s do not differ much between different users.

Our algorithms in Theorems 1 to 3 follow this framework. The differences are in the DP selection algorithms (based on central vs local model and whether we are interested in pure- or approximate-DP)—and, more importantly—how we construct the distribution $\mathcal{P}_i$ of hypotheses. In the case of the approximate-DP learner (Theorem 1), we build this on top of a learning algorithm of Ghazi et al. [GGKM21], which has a slightly stronger guarantee than that of [BNS19b]: it outputs a list of size at most $2^{\text{poly}(d)}$ with the guarantee that $h^*$ belongs to it with probability at least $1/\text{poly}(d)$. Each user runs such an algorithm $\text{poly}(d)$ times on fresh samples drawn from $\mathcal{D}$, and uses the output hypotheses to build the distribution $\mathcal{P}_i$. For pure-DP learners (Theorems 2 and 3), each user simply uses the empirical error on the probabilistic representation of the class to build the distribution $\mathcal{P}_i$.

Finally, our SQ algorithm (Theorem 5) deviates slightly from this framework. Instead of computing $\mathcal{P}_i$ outright (which is usually inefficient since its support is large), we proceed one statistical query at a time. Specifically, each user employs correlated sampling to answer each statistical query; if they manage to answer all queries in the same manner, then the algorithm will output the same hypothesis. The point here is that, since the answer to each statistical query is just a bounded-precision number in $[0, 1]$, building a probability distribution of the possible answers can be done efficiently.

## 2 Preliminaries

Let $Y = \{0, 1\}$. For a set $\Omega$, we use $2^\Omega$ to denote the set of all functions from $\Omega$ to $Y$ and use $\Delta_\Omega$ to denote the set of all distributions on $\Omega$. For distributions $\mathcal{P}, \mathcal{Q}$, we use $p \sim \mathcal{P}$ to denote that $p$ is drawn from $\mathcal{P}$ and $d_{\text{tv}}(\mathcal{P}, \mathcal{Q})$ to denote the *total variation distance* between $\mathcal{P}$ and $\mathcal{Q}$.

Let $X$ be a finite set.[2] Let $C$ denote the set of concepts from $X$ to $Y$, and let $\mathcal{D}$ be any distribution on $X \times Y$ realizable by some $h \in C$.

---

[1]Section 2.3 contains the formal definition of the selection problem and known DP algorithms for it.

[2]While our results can be extended to the case where $X$ is infinite, it does require non-trivial generalization of notation and tools (e.g., correlated sampling) to that setting.

We first recall the notion of DP. Let $\varepsilon, \delta \in \mathbb{R}_{\geq 0}$. Two datasets are *neighboring* if one can be obtained from the other by adding or removing a single user.

**Definition 6** (Differential Privacy (DP) [DMNS06, DKM$^+$06])**.** *A randomized algorithm* A *taking as input a dataset is* $(\varepsilon, \delta)$-*differentially private (*$(\varepsilon, \delta)$-*DP or* approximate-DP*) if for any two* neighboring *datasets* $D$ *and* $D'$, *and for any subset* $S$ *of outputs of* A*, it holds that* $\Pr[\mathsf{A}(D) \in S] \leq e^{\varepsilon} \cdot \Pr[\mathsf{A}(D') \in S] + \delta$*. If* $\delta = 0$*, then* A *is* $\varepsilon$-*differentially private (*$\varepsilon$-*DP or* pure-DP*).*

A dataset in our setting consists of $n$ users, where user $i$ receives a sequence of $m$ samples $(x_1^i, y_1^i), \ldots, (x_m^i, y_m^i)$ drawn i.i.d. from $\mathcal{D}$. Similar to the standard PAC setting [Val84], the algorithm A takes in the dataset and outputs a hypothesis $f$. We say that it is an $(\alpha, \beta)$-*accurate learner* if $\mathrm{err}_{\mathcal{D}}(f) \leq \alpha$ with probability $1 - \beta$ (where $\alpha, \beta \in (0, 1)$ are parameters); here, $\mathrm{err}_{\mathcal{D}}(f) = \Pr_{(x,y) \sim \mathcal{D}}[f(x) \neq y]$. We use $\mathsf{time}(\mathsf{A})$ to denote the running time of A. If A is randomized, sometimes we use the notation $\mathsf{A}(\cdot; r)$ to explicitly call out the (public) randomness $r$ it might use.

When each user holds exactly a single example (i.e., $m = 1$), we call this the *item-level* setting. We show results in both the (usual) central and local[3] models of DP.

All missing proofs are in the Supplementary Material.

## 2.1 Correlated Sampling

**Definition 7** (Correlated Sampling)**.** *A correlated sampling* *strategy for a set* $\Omega$ *with multiplicative error* $\kappa$ *is an algorithm* $\mathsf{CS} : \Delta_{\Omega} \times \mathcal{R}' \to \Omega$ *and a distribution* $\mathcal{R}'$ *on random strings such that*

- *(Marginal Correctness) For all* $\mathcal{P} \in \Delta_{\Omega}$ *and* $\omega \in \Omega$, $\Pr_{r' \sim \mathcal{R}'}[\mathsf{CS}(\mathcal{P}; r') = \omega] = \mathcal{P}(\omega)$.
- *(Error Guarantee) For* $\mathcal{P}, \mathcal{Q} \in \Delta_{\Omega}$, $\Pr_{r' \sim \mathcal{R}'}[\mathsf{CS}(\mathcal{P}; r') \neq \mathsf{CS}(\mathcal{Q}; r')] \leq \kappa \cdot d_{\mathrm{tv}}(\mathcal{P}, \mathcal{Q})$.

**Theorem 8** ([Bro97, KT02, Hol07])**.** *For any finite set* $\Omega$*, there exists a correlated sampling strategy for* $\Omega$ *with multiplicative error* 2.

## 2.2 Representation Dimension

The *size* of a hypothesis class $H$ is defined as $\mathrm{size}(H) := \log |H|$, and the size of a distribution $\mathcal{H}$ of hypothesis classes is defined as $\mathrm{size}(\mathcal{H}) := \max_{H \in \mathrm{supp}(\mathcal{H})} \mathrm{size}(H)$.

**Definition 9** (Probabilistic Representation Dimension [BNS19a])**.** *A distribution* $\mathcal{H}$ *on* $2^X$ *is said to* $(\alpha, \beta)$-*probabilistically represent* *a concept class* $C$ *if for every* $f \in C$ *and for every distribution* $\mathcal{D}$ *on* $X$*, with probability* $1 - \beta$ *over* $H \sim \mathcal{H}$*, there exists* $h \in H$ *such that* $\Pr_{x \sim \mathcal{D}}[f(x) \neq h(x)] \leq \alpha$*. The* $(\alpha, \beta)$-*probabilistic representation dimension* *of a concept class* $C$ *is defined as*

$$\mathrm{PRDim}_{\alpha, \beta}(C) := \min_{\mathcal{H} \text{ that } (\alpha, \beta)\text{-probabilistically represents } C} \mathrm{size}(\mathcal{H}).$$

*We use* $\mathrm{PRDim}(C)$ *as a shorthand for* $\mathrm{PRDim}_{1/4, 1/4}(C)$*.*

**Lemma 10** ([BNS19a])**.** *For every concept class* $C$ *and* $\alpha, \beta > 0$*, we have*

$$\mathrm{PRDim}_{\alpha, \beta}(C) \leq O\left(\log(1/\alpha) \cdot (\mathrm{PRDim}(C) + \log \log \log(1/\alpha) + \log \log(1/\beta))\right).$$

For a concept class $C$, let $\mathrm{LDim}(C)$ denote its *Littlestone dimension* [Lit87].

## 2.3 Tools from DP

In the *selection* problem, each user $i$ receives an element $u_i$ from a universe $U$. For each $u \in U$, define $c_u := |\{i \in [n] \mid u_i = u\}|$. The goal is to output $u^*$ such that $c_{u^*} \geq \max_{u \in U} c_u - \alpha$; when the output satisfies this with probability $1 - \beta$, the algorithm is said to be $(\alpha, \beta)$-*accurate*.

**Lemma 11** (Approximate-DP Selection [KKMN09, BNS19b])**.** *There is an* $(\varepsilon, \delta)$-*DP* $(O(\log(1/\delta)/\varepsilon), 0)$-*accurate algorithm for the selection problem in the central model. Moreover, the algorithm runs in* $\mathrm{poly}(n, \log |U|)$ *time.*

---

[3]A DP algorithm in the *local* model consists of a randomizer whose input is the samples held by one user and whose output is a sequence of messages, and an analyzer, whose input is the concatenation of the messages from all the randomizers and whose output is the output of the algorithm. An algorithm is DP in the local model if for any dataset, the concatenation of the outputs of all the randomizers is DP.

The following pure-DP histogram algorithm in the central model follows from the exponential mechanism [MT07]. While a trivial implementation would result in a running time that depends linearly on $|U|$, it is not hard to see that we can first toss a coin to determine whether the output would come from the input set. If so, the sampling can be done in $O(n)$ time; if not, one can randomly output one of the remaining candidates in $U$, which only requires time $O(\log |U|)$. This yields the following.

**Lemma 12** (Pure-DP Selection [MT07]). *There is an $\varepsilon$-DP $(O(\log(|U|/\beta)/\varepsilon), \beta)$-accurate algorithm for the selection problem in the central model. Moreover, the algorithm runs in $\mathrm{poly}(n, \log |U|)$ time.*

The next guarantee follows from the heavy-hitters algorithm of Bassily et al. [BNST17]:

**Lemma 13** (Pure-DP Histogram in the Local Model [EPK14]). *There is an $\varepsilon$-DP $\left(O\left(\sqrt{n \cdot \log(|U|/\beta)}/\varepsilon\right), \beta\right)$-accurate algorithm for the histogram problem in the local model. Furthermore, the algorithm runs in $\mathrm{poly}(n, \log |U|)$ time.*

# 3 Global Stability and Pseudo-Global Stability

We recall the notion of global stability of Bun et al. [BLM20] and generalize it in two ways.

**Definition 14** (Global Stability [BLM20]). *A learner A is said to be $m$-sample $\alpha$-accurate $\eta$-globally stable if there exists a hypothesis $h$ (depending on $\mathcal{D}$) such that $\mathrm{err}_{\mathcal{D}}(h) \leq \alpha$ and $\mathrm{Pr}_{(x_1,y_1),\ldots,(x_m,y_m)\sim\mathcal{D}}[\mathsf{A}((x_1,y_1),\ldots,(x_m,y_m)) = h] \geq \eta$.*

We now present the first generalization. Let $\mathcal{R}$ be a distribution of random strings.

**Definition 15** (Pseudo-Global Stability). *A learner A is said to be $m$-sample $(\alpha, \beta)$-accurate $(\eta, \nu)$-pseudo-globally stable if there exists a hypothesis $h_r$ for every $r \in \mathrm{supp}(\mathcal{R})$ (depending on $\mathcal{D}$) such that $\mathrm{Pr}_{r\sim\mathcal{R}}[\mathrm{err}_{\mathcal{D}}(h_r) \leq \alpha] \geq 1 - \beta$ and*

$$\Pr_{r\sim\mathcal{R}}\left[\Pr_{(x_1,y_1),\ldots,(x_m,y_m)\sim\mathcal{D}}[\mathsf{A}((x_1,y_1),\ldots,(x_m,y_m);r) = h_r] \geq \eta\right] \geq \nu.$$

We also generalize global stability in a slightly different manner, in order to capture the guarantees of [GGKM21].

**Definition 16** (List Global Stability). *A learner A is said to be $m$-sample $\alpha$-accurate $(L, \eta)$-list globally stable if A outputs a set of at most $L$ hypotheses and there exists a hypothesis $h$ (depending on $\mathcal{D}$) such that $\mathrm{Pr}_{(x_1,y_1),\ldots,(x_m,y_m)\sim\mathcal{D}}[h \in \mathsf{A}((x_1,y_1),\ldots,(x_m,y_m))] \geq \eta$ and $\mathrm{err}_{\mathcal{D}}(h) \leq \alpha$.*

## 3.1 Learners with Global Stability

Bun et al. [BLM20] give a globally stable learner in terms of the Littlestone dimension:

**Theorem 17** ([BLM20]). *Let $\alpha > 0$ and $C$ be any concept class with $\mathrm{LDim}(C) = d$. Then, there exists a $(2^{O(d)}/\alpha)$-sample $\alpha$-accurate $2^{-O(d)}$-globally stable learner for $C$.*

Although not explicitly stated in this manner, the improved result of Ghazi et al. [GGKM21] proceeds by giving a list globally stable learner, where the stability parameter $\eta$ is $\Omega(1/d)$, the list size $L$ is $2^{(d/\alpha)^{O(1)}}$, and the sample complexity is $(d/\alpha)^{O(1)}$.

**Theorem 18** ([GGKM21]). *Let $\alpha > 0$ and $C$ be any concept class with $\mathrm{LDim}(C) = d$. Then, there is a $(d/\alpha)^{O(1)}$-sample $\alpha$-accurate $\left(\exp\left((d/\alpha)^{O(1)}\right), \Omega(1/d)\right)$-list globally stable learner for $C$.*

We will need a slight strengthening of the above result, where there is another parameter $\zeta > 0$ and we want to ensure that every hypothesis in the output list has error at most $2\alpha$. This is stated below.

**Lemma 19.** *Let $\alpha, \zeta > 0$ and $C$ be any concept class with $\mathrm{LDim}(C) = d$. Then, there is a $(d\log(1/\zeta)/\alpha)^{O(1)}$-sample $\alpha$-accurate $\left(\exp\left((d/\alpha)^{O(1)}\right), \Omega(1/d)\right)$-list globally stable learner for $C$ such that with probability $1 - \zeta$, every hypothesis $h'$ in the output list satisfies $\mathrm{err}_{\mathcal{D}}(h') \leq 2\alpha$.*

*Proof Sketch.* This can be done by first running the algorithm in Theorem 18 to get a set $H$ of size at most $L = \exp\left((d/\alpha)^{O(1)}\right)$. Then, we draw additional $100 \cdot \log(L/\zeta)/\alpha^2$ samples $S$. Finally, we output $H' = \{h' \in H \mid \mathrm{err}_S(h') \leq 1.5\alpha\}$. By the Chernoff bound, with probability $1 - \zeta$, every hypothesis $h' \in H$ satisfies $|\mathrm{err}_\mathcal{D}(h') - \mathrm{err}_S(h')| \leq 0.5\alpha$, which yields the desired guarantees. $\qquad\square$

# 4 Approximate-DP Learner

In this section, we prove Theorem 1. We first show how to go from list global stability to pseudo-global stability using correlated sampling (Theorem 20). We then show how to go from pseudo-global stability to an approximate-DP learner using DP selection (Theorem 25).

## 4.1 From List Global Stability to Pseudo-Global Stability

**Theorem 20.** *Let $\alpha, \beta, \eta \in (0, 0.1), L \in \mathbb{N}$, and $C$ a concept class. Suppose that there exists a learner A that is $m$-sample $\alpha/2$-accurate $(L, \eta)$-list globally stable. Furthermore, with probability $1 - \left(\frac{\beta \cdot \eta^2}{10^6 \cdot \log(L/\eta)}\right)^2$, every hypothesis $h'$ in the output list satisfies $\mathrm{err}_\mathcal{D}(h') \leq \alpha$. Then, there exists a learner A' that is $m'$-sample $(\alpha, \beta)$-accurate $(1 - \beta, 1 - \beta)$-pseudo-globally stable, where $m' = O_\beta\left(m \cdot \log^3(L/\eta)/\eta^2\right)$.*

Before we prove Theorem 20, we note that with Lemma 19, it gives the following corollary.

**Corollary 21.** *Let $\alpha, \beta \in \mathbb{R}_{>0}$ and $C$ be any concept class with finite $\mathrm{LDim}(C)$. Then, there exists a learner A that is $m$-sample $(\alpha, \beta)$-accurate and $(1 - \beta, 1 - \beta)$-pseudo-globally stable, where $m = O_\beta((\mathrm{LDim}(C)/\alpha)^{O(1)})$.*

*Proof of Theorem 20.* Let $\tau = 0.5\eta, \gamma = \frac{10^6 \log(L/(\beta\tau))}{\tau}, k_1 = \frac{10^6 \log(L/(\beta\tau))}{\tau^2}$, and $k_2 = \lceil \frac{10^6 \gamma^2 \cdot \log(L/(\tau\beta))}{\beta^4} \rceil$. Let $\mathsf{CS}$ be a correlated sampling strategy for $2^X$ and let $\mathcal{R}'$ be the (public) randomness it uses, as in Theorem 8. Algorithm 1 presents our learner A'.

---

**Algorithm 1** Pseudo-Globally Stable Learner A'.

    **for** $i = 1, \ldots, k_1$ **do**
        Draw $S_i \sim \mathcal{D}^m$, run A on $S_i$ to get a set $H_i$
    Let $H$ be the set of all $f \in 2^X$ that appears in at least $\tau \cdot k_1$ of the sets $H_1, \ldots, H_{k_1}$
    **for** $j = 1, \ldots, k_2$ **do**
        Draw $T_j \sim \mathcal{D}^m$, run A on $T_j$ to get a set $G_j$
    **for** $h \in H$ **do**
        Let $\hat{Q}_{H, G_1, \ldots, G_{k_2}}(h) = \frac{|\{j \in [k_2] \mid h \in G_j\}|}{k_2}$
    Let $\hat{\mathcal{P}}_{H, G_1, \ldots, G_{k_2}}$ be the probability distribution on $2^X$ defined by

$$\hat{\mathcal{P}}_{H, G_1, \ldots, G_{k_2}}(h) = \begin{cases} \frac{\exp(\gamma \cdot \hat{Q}_{G_1, \ldots, G_{k_2}}(h))}{\sum_{h' \in H} \exp(\gamma \cdot \hat{Q}_{G_1, \ldots, G_{k_2}}(h'))} & \text{if } h \in H, \\ 0 & \text{otherwise.} \end{cases}$$

    Output $\mathsf{CS}(\hat{\mathcal{P}}_{H, G_1, \ldots, G_j}; r')$, where $r' \sim \mathcal{R}'$

---

Notice that the number of samples used in A' is $m \cdot (k_1 + k_2) = m \cdot O_\beta(\log^3(L/\eta)/\tau^2)$ as claimed.

**(Accuracy Analysis)** Since we assume that the output of A consists only of hypotheses with distributional error at most $\alpha$ with probability $1 - \beta/k_1$, a union bound implies that this holds for all hypotheses in $H$ with probability $1 - \beta$. This yields the desired $(\alpha, \beta)$-accuracy of the algorithm.

**(Pseudo-Global Stability Analysis)** For this, we need a few additional notation. First, for every $h \in 2^X$, we let $Q(h)$ denote $\Pr_{S \sim \mathcal{D}^m}[h \in A(S)]$. Moreover, let $H_{\geq 1.1\tau} = \{h \in 2^X \mid Q(h) \geq 1.1\tau\}$ and similarly $H_{\geq 0.9\tau} = \{h \in 2^X \mid Q(h) \geq 0.9\tau\}$. A crucial property we will use is that $H$ is w.h.p. sandwiched between $H_{\geq 1.1\tau}$ and $H_{\geq 0.9\tau}$, as stated below.

**Lemma 22.** *Let $\mathscr{E}$ denote the event that $H_{\geq 1.1\tau} \subseteq H \subseteq H_{\geq 0.9\tau}$. Then,*

$$\Pr[\mathscr{E}] \geq 1 - \beta^2/30,$$

*where the probability is over the randomness of $S_1, \ldots, S_{k_1}$ and that of $\mathsf{A}$ on these datasets.*

*Proof of Lemma 22.* We will separately argue that $\Pr[H_{\geq 1.1\tau} \not\subseteq H] \leq \beta^2/60$ and $\Pr[H \not\subseteq H_{\geq 0.9\tau}] \leq \beta^2/60$. A union bound then yields the claimed statement.

To prove the first bound, observe that since $\mathsf{A}$ outputs a set of size at most $L$, $|H_{\geq 1.1\tau}| \leq L/(1.1\tau) < L/\tau$. Consider each $f \in H_{\geq 1.1\tau}$; notice that $\mathbf{1}[f \in H_i]$ is simply an i.i.d. Bernoulli random variable with success probability $Q(f) \geq 1.1\tau$. Hence, by the Hoeffding inequality, we have

$$\Pr[f \notin H] \leq \exp\left(-0.02\tau^2 k_1\right) < 0.001\beta^2\tau/L,$$

where the last inequality follows from our choice of $\tau, k_1$. Taking a union bound over all $f \in H_{\geq 1.1\tau}$ concludes our proof for the first inequality.

For the second inequality, consider the set $H_{<0.9\tau} := 2^X \setminus H_{\geq 0.9\tau}$. Since each element $f \in H_{<0.9\tau}$ satisfies $Q(f) < 0.9\tau$, we may partition[4] $H_{<0.9\tau}$ into $H_{<0.9\tau}^1 \cup \cdots \cup H_{0.9\tau}^q$ such that $\sum_{f \in H_{<0.9\tau}^j} Q(f) < 0.9\tau$ for all $j \in [q]$ and $q \leq L/(0.45\tau) + 1 < 4L/\tau$. Fix $j \in [q]$; notice that

$$\Pr[H \cap H_{<0.9\tau}^j \neq \emptyset] \leq \Pr[|\{i \in [k_1] \mid H_i \cap H_{<0.9\tau}^j \neq \emptyset\}| \geq \tau k_1].$$

Now, each $\mathbf{1}[H_i \cap H_{<0.9\tau}^j \neq \emptyset]$ is an i.i.d. Bernoulli random variable with success probability at most $\sum_{f \in H_{<0.9\tau}^j} Q(f) < 0.9\tau$. Thus, we can apply the Hoeffding inequality to conclude that

$$\Pr[H \cap H_{<0.9\tau}^j \neq \emptyset] \leq \exp(-0.02\tau^2 k_1) < 0.001\beta^2\tau/L.$$

Taking a union bound over all $j \in [q]$, we have $\Pr[H \cap H_{<0.9\tau} \neq \emptyset] < 0.01\beta^2$. This completes our proof of the second inequality. $\qquad\square$

Next, let $\mathcal{P}$ be the probability distribution on $2^X$ defined by

$$\mathcal{P}(f) = \begin{cases} \frac{\exp(\gamma \cdot Q(f))}{\sum_{f' \in H_{\geq 0.9\tau}} \exp(\gamma \cdot Q(f'))} & \text{if } f \in H_{\geq 0.9\tau} \\ 0 & \text{otherwise.} \end{cases}$$

Furthermore, let $\mathcal{P}_H$ be the probability distribution on $2^X$ defined by

$$\mathcal{P}_H(f) = \begin{cases} \frac{\exp(\gamma \cdot Q(f))}{\sum_{f' \in H} \exp(\gamma \cdot Q(f'))} & \text{if } f \in H, \\ 0 & \text{otherwise.} \end{cases}$$

Once again, notice that $\mathcal{P}$ is independent of the run of the algorithm (i.e., it only depends on $\mathsf{A}$), whereas $\mathcal{P}_H$ can vary on different runs, depending on $H$. Our first component of the proof is to argue that $\mathcal{P}$ and $\mathcal{P}_H$ are often close:

**Lemma 23.** *When $\mathscr{E}$ holds, we have $d_{\mathrm{tv}}(\mathcal{P}, \mathcal{P}_H) \leq \beta^2/30$.*

*Proof of Lemma 23.* Recall that from the assumption of list global stability, there exists $h$ such that $Q(h) \geq \eta = 2\tau$. When $\mathscr{E}$ holds, $H$ is a subset of $H_{\geq 0.9\tau}$, meaning that $\mathcal{P}_H$ is the conditional probability of $\mathcal{P}$ on $H$. Thus, we have

$$d_{\mathrm{tv}}(\mathcal{P}, \mathcal{P}_H) \leq \mathcal{P}(H \setminus H_{\geq 0.9\tau}) = \frac{\sum_{f \in H_{\geq 0.9\tau} \setminus H} \exp(\gamma \cdot Q(f))}{\sum_{f' \in H_{\geq 0.9\tau}} \exp(\gamma \cdot Q(f'))} \overset{(a)}{\leq} \frac{\sum_{f \in H_{\geq 0.9\tau} \setminus H_{\geq 1.1\tau}} \exp(\gamma \cdot Q(f))}{\sum_{f' \in H_{\geq 0.9\tau}} \exp(\gamma \cdot Q(f'))}$$

$$\leq \frac{\sum_{f \in H_{\geq 0.9\tau} \setminus H_{\geq 1.1\tau}} \exp(\gamma \cdot 1.1\tau)}{\sum_{f' \in H_{\geq 0.9\tau}} \exp(\gamma \cdot Q(f'))} \leq \frac{|H_{\geq 0.9\tau}| \cdot \exp(\gamma \cdot 1.1\tau)}{\exp(\gamma \cdot Q(h))}$$

$$\overset{(b)}{\leq} |H_{\geq 0.9\tau}| \cdot \exp(-\gamma \cdot 0.9\tau) \leq L/(0.9\tau) \cdot \exp(-\gamma \cdot 0.9\tau) \overset{(c)}{\leq} \beta^2/30,$$

where inequality (a) follows from $H_{\geq 1.1\tau} \subseteq H$ (which in turns holds because of $\mathscr{E}$), inequality (b) follows since $Q(h) \geq 2\tau$, and inequality (c) follows from our choice of $\gamma$. $\qquad\square$

---

[4] A simple way is to start with a singleton partition and then merge any two parts whose total $Q(\cdot)$ is less than $0.9\tau$; in the end, we will left with a partition where all but at most one part has weight at least $0.45\tau$.

Next, we show that $\mathcal{P}_H$ is often close to its "empirical" version $\hat{\mathcal{P}}_{H,G_1,\ldots,G_{k_2}}$.

**Lemma 24.** $\mathbb{E}[d_{\mathrm{tv}}(\mathcal{P}_H, \hat{\mathcal{P}}_{H,G_1,\ldots,G_{k_2}})] \le \beta^2/30$ *where the probability is over the randomness of* $T_1,\ldots,T_{k_2}$ *and that of* A*'s executions on these datasets.*

*Proof of Lemma 24.* From how $H$ is selected and from the assumption that the output of A has size at most $L$, we have $|H| \le L/\tau$. Now, fix $f \in H$. Note that $\hat{Q}_{H,G_1,\ldots,G_{k_2}}(f)$ is simply an average of $k_2$ i.i.d. Bernoulli random variables with success probability $Q(f)$. Using the Hoeffding inequality,

$$\Pr\left[|\hat{Q}_{H,G_1,\ldots,G_{k_2}}(f) - Q(f)| > 100\sqrt{\frac{\log(L/(\tau\beta))}{k_2}}\right] \le \frac{\beta^2}{60 \cdot (L/\tau)}. \tag{1}$$

By a union bound over all $f \in H$, we can conclude that with probability $1 - \beta^2/60$ we have $|\hat{Q}_{H,G_1,\ldots,G_{k_2}}(f) - Q(f)| \le 100\sqrt{\frac{\log(L/(\tau\beta))}{k_2}}$ for all $f \in H$. When this holds, we have

$$d_{\mathrm{tv}}(\mathcal{P}_H, \hat{\mathcal{P}}_{H,G_1,\ldots,G_{k_2}}) = \sum_{h \in h} \mathcal{P}_H(h) \cdot \max\left\{0, \left(\frac{\hat{\mathcal{P}}_{H,G_1,\ldots,G_{k_2}}(h)}{\mathcal{P}_H(h)} - 1\right)\right\}$$

$$\overset{(1)}{\le} \sum_{h \in h} \mathcal{P}_H(h) \cdot \left(\exp\left(\gamma \cdot 100\sqrt{\frac{\log(L/(\tau\beta))}{k_2}}\right) - 1\right) \le \exp\left(\gamma \cdot 100\sqrt{\frac{\log(L/(\tau\beta))}{k_2}}\right) - 1$$

$$\le 200\gamma\sqrt{\frac{\log(L/(\tau\beta))}{k_2}} \le \beta^2/60, \text{ from our choice of } k_2. \qquad \square$$

Combining Lemmas 22 to 24, we can conclude that $\mathbb{E}[d_{\mathrm{tv}}(\mathcal{P}, \hat{\mathcal{P}}_{H,G_1,\ldots,G_{k_2}})] \le \beta^2/10$ where the expectation is over all the randomness involved in the algorithm except $r'$. Now, let $h_{r'} = \mathsf{CS}(P; r')$. From the error guarantee of correlated sampling in Theorem 8, we can conclude that

$$\Pr_{r' \sim \mathcal{R}', S_1,\ldots,S_{k_1}, T_1,\ldots,T_{k_2}}[\mathsf{A}'(S_1,\ldots,S_{k_1}, T_1,\ldots,T_{k_2}; r') \ne h_{r'}] \le \beta^2/5.$$

Thus, applying Markov inequality, we get the pseudo-global stability of A

$$\Pr_{r' \sim \mathcal{R}'}\left[\Pr_{S_1,\ldots,S_{k_1}, T_1,\ldots,T_{k_2}}[\mathsf{A}'(S_1,\ldots,S_{k_1}, T_1,\ldots,T_{k_2}; r') = h_{r'}] \ge 1 - \beta\right] \ge 1 - \beta. \qquad \square$$

## 4.2 From Pseudo-Global Stability to Approximate-DP Learner in the Central Model

In this section we prove Theorem 25 that allows us to convert any pseudo-globally stable learner to an approximate-DP learner. Combining Theorem 25 and Corollary 21 yields Theorem 1.

**Theorem 25.** *Let* $\alpha, \beta \in (0, 0.1)$ *and* $C$ *a concept class. Suppose that there exists a learner* A *that is* $m$-sample $(\alpha, \beta/3)$-accurate $(0.9, 1 - \beta/3)$-pseudo-globally stable. Then, for any $\varepsilon \in (0,1)$, there is an $(\varepsilon, \delta)$-DP $(\alpha, \beta)$-accurate learner for $C$ in the central model that requires $n = O\left(\log(1/(\delta\beta))/\varepsilon\right)$ users where each user has $m$ samples. The learner runs in time $\mathrm{poly}(\mathsf{time}(\mathsf{A}), n, d)$.*

*Proof.* Let $n = K \cdot \log\left(1/(\delta\beta)\right)/\varepsilon$ where $K$ is a sufficiently large constant. Let $\mathcal{R}$ denote the public randomness shared between the users. Algorithm 2 shows our approximate-DP learner.

---
**Algorithm 2** Approximate-DP Learner in the Central Model.

---
User $i$ randomly draws $m$ samples $S_i \sim \mathcal{D}^m$ and runs $\mathsf{A}(S_i; r)$ to obtain a hypothesis $h_i$
Run the $(\varepsilon, \delta)$-DP selection algorithm from Lemma 11 where $U = 2^X$ and user $i$'s item is $h_i$
Return the output $h^*$ from the previous step

---

It is clear that Algorithm 2 is $(\varepsilon, \delta)$-DP. We will now analyze its accuracy. First, from the definition of pseudo-global stability and a union bound, with probability $1 - 2\beta/3$ over $r \sim \mathcal{R}$, there exists $h_r$

such that $\mathrm{err}_{\mathcal{D}}(h_r) \leq \alpha$, and $\Pr_{S \sim \mathcal{D}^m}[\mathsf{A}(S; r) = h_r] \geq 0.9$. Conditioned on this, we can use the Hoeffding inequality to conclude that, with probability $1 - \beta/6$, we have[5]

$$c_{h_r} \geq 0.8n. \tag{2}$$

Conditioned on (2), Lemma 11 guarantees that $c_{h^*} \geq \max_{h \in 2^X} c_h - 0.1n$ with probability $1 - \beta/6$ (for sufficiently large $K$); when this is the case, Algorithm 2 outputs $h^* = h_r$. Hence, applying a union bound (over this and (2)), we can conclude that the algorithm is $(\alpha, \beta)$-accurate as desired. $\square$

## 5    Conclusions and Future Directions

In this work, we study the question of learning with user-level DP when each user may have many examples. We prove tight upper and lower bounds on the *number of users* required to learn each concept class, provided that each user has sufficiently many i.i.d. samples. An immediate open question here is whether one can also derive a tight bound on the *number of samples per users* required; note that this bound will depend on the number of users. For approximate-DP learning, this problem might be hard because the big gap in the item-level learning setting between $\mathrm{poly}(\mathrm{LDim})$ [GGKM21] and $\Omega(\log^* \mathrm{LDim})$ [ALMM19]) is still open. The pure-DP case might be easier since tight bounds are known both in the central [BNS19a] and the local models (up to a polynomial factor) [KLN+11].

Another interesting direction is to derive additional efficient user-level DP learners whose sample complexities are better than item-level DP learners. We give two "weak" examples of this in Appendix B for the case of the non-interactive local model. It would be good to give such an example for general algorithms in the local model as well. On this front, PARITY seems to be a good candidate; as stated earlier, it has SQ dimension $2^{\Omega(d)}$ [BFJ+94] meaning that it requires $2^{\Omega(d)}$ samples in the (interactive) item-level local model [KLN+11]. Can we come up with an efficient user-level DP algorithm in the local model that requires $\mathrm{poly}(d)$ samples in total?

Furthermore, our algorithms make extensive use of shared randomness—in the form of correlated sampling. Is this necessary? In particular,

- Is there a local user-level DP algorithm for learning any class $C$ using $\mathrm{poly}(\mathrm{PRDim}(C))$ users each having $\mathrm{poly}(\mathrm{PRDim}(C))$ samples, without using public randomness? In other words, can the use of public randomness be removed from Theorem 2?
- Is there an $\eta$-globally stable learner for any class $C$ with finite Littlestone dimension where $\eta > 0$ is some absolute constant? In other words, can the use of public randomness be removed from Corollary 21?

While it is not hard to show that the second question has a negative answer if we require $\eta > 1/2$, we are not aware of a proof that $\eta$ must go to zero for some family of concept classes.

Lastly, it would also be interesting to see whether techniques employed in our paper may be useful beyond the PAC setting. For example, Golowich [Gol21] gives DP regression algorithms based on stability notions similar to [BLM20, GGKM21] and it is plausible that our approach gives user-level regression algorithm in a setting similar to [Gol21].

### Acknowledgement

We thank Jessica Sorrell for pointing us to [ILPS21] and explaining the main results of that paper.

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
