# A  Pure-DP Learners

In this section we prove Theorem 2 and Theorem 3. First we show how to go from probabilistic representation to pseudo-global stability (Theorem 26) using correlated sampling. Next we show how to go from pseudo-global stability to a pure-DP learner using the histogram algorithm in the local (Theorem 29) model and the central (Theorem 30) model.

## A.1  Pseudo-Globally Stable Learner from Probabilistic Representation

**Theorem 26.** *Let $\alpha, \beta \in \mathbb{R}_{>0}$ and $C$ be any concept class with finite $\mathrm{PRDim}(C)$. Then, there exists a learner* A *that is $m$-sample $(\alpha, \beta)$-accurate and $(1 - \beta, 1 - \beta)$-pseudo-globally stable, where $m = O_\beta\left(\mathrm{PRDim}_{\alpha/2, \beta/2}(C)^3 / \alpha^2\right)$. Furthermore, the public randomness specifies $H \in \mathrm{supp}(\mathcal{H})$ such that the output of* A *belongs to $H$.*

Before we prove Theorem 26, we note that this along with Lemma 10 gives the following corollary.

**Corollary 27.** *Let $\alpha, \beta \in \mathbb{R}_{>0}$ and $C$ be any concept class with finite $\mathrm{PRDim}(C)$. Then, there exists a learner* A *that is $m$-sample $(\alpha, \beta)$-accurate and $(1 - \beta, 1 - \beta)$-pseudo-globally stable, where $m = \tilde{O}_\beta\left(\mathrm{PRDim}(C)^3 / \alpha^2\right)$. Furthermore, the public randomness specifies $H \in \mathrm{supp}(\mathcal{H})$ such that the output of* A *belongs to $H$.*

*Proof of Theorem 26.* For brevity, let $d := \mathrm{PRDim}_{\alpha/2, \beta/2}(C)$ and $\gamma = \frac{2(d + \log(1/\beta) + 10)}{\alpha}$. Let CS be a correlated sampling strategy for $2^X$ and let $\mathcal{R}'$ be the (public) randomness it uses, as in Theorem 8 and let $\mathcal{H}$ be an $(\alpha/2, \beta/2)$-probabilistic representation of $C$ such that $\mathrm{size}(\mathcal{H}) = d$.

The public randomness used in our learner A is split into two parts: $r' \sim \mathcal{R}'$ and $H \sim \mathcal{H}$. Algorithm 3 presents the pseudo-globally stable learner A.

---
**Algorithm 3** Pseudo-Globally Stable Learner A.

---
Draw $m := \frac{10^6 \gamma^2 d \log(1/\beta)}{\beta^4}$ samples from $\mathcal{D}$; let $S$ denote the multiset of these samples

Let $\hat{\mathcal{P}}_{H,S}$ denote the probability distribution on $\Delta_{2^X}$ where

$$\hat{\mathcal{P}}_{H,S}(h) = \begin{cases} \frac{\exp(-\gamma \cdot \mathrm{err}_S(h))}{\sum_{h' \in H} \exp(-\gamma \cdot \mathrm{err}_S(h'))} & h \in H, \\ 0 & \text{otherwise.} \end{cases}$$

Output $\mathrm{CS}(\hat{\mathcal{P}}_{H,S}; r')$

---

We remark that the last property in the theorem statement holds simply because $H$ is part of the public randomness and A always outputs a hypothesis that belongs to $H$.

To prove the pseudo-global stability and accuracy of the learner, we introduce additional notation: Let $\mathcal{P}_H$ denote the probability distribution on $\Delta_{2^X}$ where

$$\mathcal{P}_H(h) = \begin{cases} \frac{\exp(-\gamma \cdot \mathrm{err}_{\mathcal{D}}(h))}{\sum_{h' \in H} \exp(-\gamma \cdot \mathrm{err}_{\mathcal{D}}(h'))} & h \in H, \\ 0 & \text{otherwise.} \end{cases}$$

In other words, $\mathcal{P}_H$ is the distributional version of (the empirical) $\hat{\mathcal{P}}_{H,S}$.

**(Pseudo-Global Stability Analysis)**  To show $(1 - \beta, 1 - \beta)$-pseudo-global stability, we start by bounding the total variation distance between $\mathcal{P}_H$ and $\hat{\mathcal{P}}_{H,S}$.

**Lemma 28.** *For a fixed $H \in \mathrm{supp}(\mathcal{H})$, we have*

$$\Pr_{S \sim \mathcal{D}^m}[d_{\mathrm{tv}}(\mathcal{P}_H, \hat{\mathcal{P}}_{H,S}) \leq \beta/10] \geq 1 - \beta^2/10. \tag{3}$$

*Proof of Lemma 28.* By the Hoeffding inequality, for each $h \in \mathcal{H}$, we have $\Pr_{S \sim \mathcal{D}^m}[|\mathrm{err}_{\mathcal{D}}(h) - \mathrm{err}_S(h)| \leq 10\sqrt{d \log(1/\beta)/m}] \leq \frac{\beta^2}{10 \cdot 2^d}$. By a union bound, with probability $1 - \beta^2/10$ we have

$|\operatorname{err}_{\mathcal{D}}(h) - \operatorname{err}_S(h)| \leq 10\sqrt{d\log(1/\beta)/m}$ for all $h \in H$. When this holds, we have

$$d_{\mathrm{tv}}(\mathcal{P}_H, \hat{\mathcal{P}}_{H,S}) = \sum_{h \in h} \mathcal{P}_H(h) \cdot \max\left\{0, \left(\frac{\hat{\mathcal{P}}_{H,S}(h)}{\mathcal{P}_H(h)} - 1\right)\right\}$$

$$\leq \sum_{h \in h} \mathcal{P}_H(h) \cdot \left(\exp\left(\gamma \cdot 20\sqrt{d\log(1/\beta)/m}\right) - 1\right)$$

$$\leq \exp\left(\gamma \cdot 20\sqrt{d\log(1/\beta)/m}\right) - 1$$

$$\leq 40\gamma\sqrt{d\log(1/\beta)/m}$$

(From our choice of $m$) $\leq \beta^2/10$. $\qquad\square$

Let $h_{(r',H)} = \mathsf{CS}(\mathcal{P}_H; r')$. Recall from the guarantee of the correlated sampling algorithm that

$$\Pr_{r' \sim \mathcal{R}'}[\mathsf{CS}(\hat{\mathcal{P}}_{H,S}; r') \neq h_{(r',H)}] \leq 2 \cdot d_{\mathrm{tv}}(\mathcal{P}_H, \hat{\mathcal{P}}_{H,S}).$$

Combining the above inequality with (3), for any fix $H \in \mathrm{supp}(\mathcal{H})$, we have

$$\Pr_{r' \sim \mathcal{R}'} \Pr_{S \sim \mathcal{D}^m}[\mathsf{A}(S) \neq h_{(r',H)}] \leq \mathbb{E}_{d_{\mathrm{tv}}}[2 \cdot d_{\mathrm{tv}}(\mathcal{P}_H, \hat{\mathcal{P}}_{H,S})] \leq 2\beta^2/5 < \beta^2.$$

From this, we can conclude that

$$\Pr_{r' \sim \mathcal{R}'}\left[\Pr_{S \sim \mathcal{D}^m}[\mathsf{A}(S) = h_{(r',H)}] \geq 1 - \beta\right] \geq 1 - \beta,$$

for all $H \in \mathrm{supp}(\mathcal{H})$.

**(Accuracy Analysis)** First, recall from the definition of $(\alpha/2, \beta/2)$-probabilistic representation of $C$, with probability $1 - \beta/2$ over $H \sim \mathcal{H}$, we have

$$\exists h^* \in H, \operatorname{err}_{\mathcal{D}}(h^*) \leq \alpha/2. \tag{4}$$

When this holds, we can analyze as in [HT10] for the accuracy of the exponential mechanism. More formally, from the marginal correctness of the correlated sampling algorithm, we have

$$\Pr_{r' \sim \mathcal{R}'}[\operatorname{err}_{\mathcal{D}}(h_{r',H}) > \alpha] = \Pr_{h \sim P_H}[\operatorname{err}_{\mathcal{D}}(h) > \alpha]$$

$$= \frac{\sum_{\substack{h \in H \\ \operatorname{err}_{\mathcal{D}}(h) > \alpha}} \exp(-\gamma \cdot \operatorname{err}_{\mathcal{D}}(h))}{\sum_{h' \in H} \exp(-\gamma \cdot \operatorname{err}_{\mathcal{D}}(h'))}$$

$$\leq \frac{|\mathcal{H}| \cdot \exp(-\gamma \cdot \alpha)}{\exp(-\gamma \cdot \operatorname{err}_{\mathcal{D}}(h^*))}$$

(From (4)) $\leq 2^d \cdot \exp(-\gamma \cdot \alpha/2)$

(From our choice of $\gamma$) $\leq \beta/2$.

Combining the above inequality with the fact that (4) holds with probability at least $1 - \beta/2$,

$$\Pr_{r' \sim \mathcal{R}', H \sim \mathcal{H}}[\operatorname{err}_{\mathcal{D}}(h_{r',H}) > \alpha] \geq 1 - \beta,$$

which concludes our proof. $\qquad\square$

### A.2 From Pseudo-Globally Stable Learner to Pure-DP Learner in the Local Model

In this section we prove Theorem 29, which allows us to convert any pseudo-globally stable learner to a pure-DP learner in the local model. Combining this and Corollary 27 yields Theorem 2.

**Theorem 29.** *Let $\alpha, \beta \in (0, 0.1)$ and $C$ a concept class. Suppose that there exists a learner* A *that is $m$-sample $(\alpha, \beta/3)$-accurate and $(0.9, 1 - \beta/3)$-pseudo-globally stable. Furthermore, suppose that each public randomness $r$ of* A *specifies a hypothesis class $H_r$ of size at most $d$ such that* A *outputs (on that public randomness) always belong to $H_r$. Then, for any $\varepsilon \in (0, 1)$, there exists an $\varepsilon$-DP $(\alpha, \beta)$-accurate learner in the (public randomness) local model for $C$ that requires $n = O\left(\frac{d + \log(1/\beta)}{\varepsilon^2}\right)$ users where each user has $m$ samples. Moreover, the running time of the learner is $\mathrm{poly}(\mathrm{time}(\mathsf{A}), n, d)$.*

*Proof.* Let $n = K\left(d + \log(1/\beta)\right)/\varepsilon^2$ where $K$ is a sufficiently large constant. Let $r$ denote the public randomness shared between the users. Algorithm 4 presents our pure-DP learner.

---

**Algorithm 4** Pure-DP Learner in the Local Model.

---

User $i$ draws $m$ samples $S_i \sim \mathcal{D}^m$
User $i$ runs $\mathsf{A}(S_i; r)$ to obtain hypothesis $h_i \in H_r$
Run the $\varepsilon$-DP selection algorithm in the local model from Lemma 13, where $U = H_r$ and user $i$'s item is $h_i$.
Return the output $h^*$ from the previous step

---

It is obvious to see that the algorithm is $\varepsilon$-DP in the local model. We will now analyze its accuracy. First, from definition of pseudo-global stability and a union bound, with probability $1 - 2\beta/3$ over $r \sim \mathcal{R}$ there exists $h_r$ such that $\mathrm{err}_{\mathcal{D}}(h_r) \leq \alpha$, and $\Pr_{S \sim \mathcal{D}^m}[\mathsf{A}(S; r) = h_r] \geq 0.9$. Conditioned on this, we can use the Hoeffding inequality to conclude that, with probability $1 - \beta/6$,

$$c_{h_r} \geq 0.8. \tag{5}$$

Conditioned on (5), we can use Lemma 13 to guarantee that $c_{h^*} \geq \max_{h \in 2^X} c_h - 0.1n$ with probability at least $1 - \beta/6$ (when $K$ is sufficiently large); when this is the case, the algorithm outputs $h^* = h_r$. Hence, applying a union bound (over this and (5)), we can conclude that the algorithm outputs a hypothesis with error at most $\alpha$ with respect to $\mathcal{D}$ as desired. $\square$

### A.3 From Pseudo-Globally Stable Learner to Pure-DP Learner in the Central Model

We next prove the following result, which is similar to Theorem 29 except that we now work in the central model and achieve a slightly better bound, i.e., the dependency of the number of users on $\varepsilon$ is $1/\varepsilon$ instead of $1/\varepsilon^2$. Plugging Theorem 29 into Corollary 27, we get Theorem 3.

**Theorem 30.** *Let $\alpha, \beta \in (0, 0.1)$ and $C$ a concept class. Suppose that there exists a learner $\mathsf{A}$ that is $m$-sample $(\alpha, \beta/3)$-accurate and $(0.9, 1 - \beta/3)$-pseudo-globally stable. Furthermore, suppose that each public randomness $r$ of $\mathsf{A}$ specifies a hypothesis class $H_r$ of size at most $d$ such that $\mathsf{A}$ outputs (on that public randomness) always belong to $H_r$. Then, for any $\varepsilon \in (0, 1)$, there exists an $\varepsilon$-DP $(\alpha, \beta)$-accurate learner in the central model for $C$ that requires $n = O\left(\frac{d + \log(1/\beta)}{\varepsilon}\right)$ users where each user has $m$ samples. Moreover, the running time of the learner is $\mathrm{poly}(\mathsf{time}(\mathsf{A}), n, d)$.*

The proof of Theorem 30 is essentially the same as that of Theorem 29, except that we are using the histogram algorithm in the central model (Lemma 12) instead of in the local model (Lemma 13).

## B Efficient Reduction for SQ Algorithms

This section is devoted to the proof of Theorem 5. To understand the intuition behind the proof, observe that our algorithms from the previous section are inefficient because they have to estimate a certain probability distribution over the set of possible output hypotheses (on which a correlating sampling is then applied); this set can be large, resulting in the inefficiency. To overcome this, we observe that in the SQ model our job is now to simply produce a single number—the output of the oracle—which is then returned back to $\mathsf{A}$. Since it is a single number (and can have error as large as $\tau$), we can quite easily estimate its value and use correlated sampling to round it to some nearby number. By doing this for every oracle call from $\mathsf{A}$, we can obtain a pseudo-globally stable algorithm, which can then be turned into user-level DP algorithms using Theorems 25, 29 and 30.

### B.1 From SQ Algorithms to Pseudo-Globally Stable Learners

We recall the definition of statistical queries [Kea98].

**Definition 31** (Statistical Query Oracle [Kea98])**.** *For a given distribution $\mathcal{D}$ and accuracy parameter $\tau > 0$, a statistical query (SQ) $\mathrm{STAT}_{\mathcal{D}}(\tau)$ is an oracle that, when given a function $\phi : \mathrm{supp}(\mathcal{D}) \to [-1, 1]$, outputs some number $o$ such that $|o - \mathbb{E}_{z \sim \mathcal{D}}[\phi(z)]| \leq \tau$.*

For simplicity, we only present the proof for the case where A is deterministic. The randomized case can be handled similarly, by additionally using public randomness as A's private randomness.

**Lemma 32.** *Let $C$ be any concept class, and suppose that there exists an algorithm* A *that can $\alpha$-learn $C$ using $q$ $\text{STAT}_\mathcal{D}(\tau)$ queries. Then, there exists a learner* A' *that is $m$-sample $(\alpha, \beta)$-accurate $(1 - \beta, 1 - \beta)$-pseudo-globally stable, where $m = \tilde{O}_\beta\left(q^3/\tau^2\right)$. Furthermore, the running time of* A' *is at most* $\text{poly}(\text{time}(\mathsf{A}), 1/\tau, 1/\beta)$.

Plugging Lemma 32 into Theorems 25, 29 and 30 implies Theorem 5.

*Proof of Lemma 32.* Let $I = \lceil 3/\tau \rceil$, $\Omega = \{0, 1/I, \ldots, (I-1)/I, 1\}$ and $m' = 10^6 \cdot \frac{q^2}{\beta^2 \tau^2} \cdot \log\left(\frac{q^2}{\beta^4 \tau^2}\right)$. Let CS with randomness $\mathcal{R}'$ be the correlated sampling strategy for $\Omega$ given in Theorem 8. Let $\mathcal{R} = (\mathcal{R}')^{\otimes q}$.

For $r = (r'_1, \ldots, r'_q) \sim \mathcal{R}$, we simulate the oracle $\text{STAT}_\mathcal{D}(\tau)$ as follows:

---

**Algorithm 5** Pseudo-Globally Stable Learner from SQ.

---

**for** the $i$th query $\phi_i$ to the SQ oracle **do**

    Draw $m'$ samples $S_i \sim \mathcal{D}^{m'}$

    $\hat{u}_i \leftarrow \frac{1}{m's} \sum_{z \in S_i} \phi_i(z)$

    Let $\hat{\mathcal{P}}_i$ denote the probability distribution on $\Omega$ where

$$\hat{\mathcal{P}}_i(\ell/I) = \begin{cases} \hat{u}_i - \lfloor \hat{u}_i \rfloor & \text{if } \ell = \lfloor \hat{u}_i \rfloor + 1, \\ 1 - (\hat{u}_i - \lfloor \hat{u}_i \rfloor) & \text{if } \ell = \lfloor \hat{u}_i \rfloor, \\ 0 & \text{otherwise.} \end{cases}$$

    Output $\text{CS}(\hat{\mathcal{P}}_i; r'_i)$

---

Let A' denote the learner that runs A using the above oracle simulation. It worth keeping in mind that, when A is adaptive, $\hat{\mathcal{P}}_i$ depends on all of $S_1, r'_1, \ldots, S_{i-1}, r'_{i-1}, S_i$; however, we do note write this explicitly for notational ease.

Now, consider the "distributional" runs of the algorithm A where the oracle is as defined above except the "empirical" $\hat{u}_i$ is replaced by the "distributional" $u_i := \mathbb{E}_{z \sim \mathcal{D}}[\phi_i(z)]$, and similarly $\hat{P}_i$ is replaced by $\mathcal{P}_i$ where

$$\mathcal{P}_i(\ell/I) = \begin{cases} u_i - \lfloor u_i \rfloor & \text{if } \ell = \lfloor u_i \rfloor + 1, \\ 1 - (u_i - \lfloor u_i \rfloor) & \text{if } \ell = \lfloor u_i \rfloor, \\ 0 & \text{otherwise,} \end{cases}$$

and the output of the "distributional" oracle is now $\text{CS}(\mathcal{P}_i; r'_i)$. Let $h_r$ denote the output of the learner A when using this "distributional" oracle. Notice that these outputs depend only on the distribution $\mathcal{D}$ and the randomness $r$.

Let $\mathscr{E}_i$ denote the event that the output of oracle for the $i$th query is the same in the two cases. We will show that

$$\Pr_{S_i, r'_i}[\mathscr{E}_i \mid \mathscr{E}_{i-1}, \ldots, \mathscr{E}_1] \geq 1 - \beta^2/q. \tag{6}$$

Before we prove (6), let us first explain why it implies our proof. By a union bound, (6) implies that, with probability $1 - \beta^2$ (over $S_1, \ldots, S_q, r$), the "empirical" version of the oracle answers the same $q$ queries as the "distributional" version, meaning that $\mathcal{A}$ will return the same output in the former as in the latter; more formally, we have

$$\Pr_{r \sim \mathcal{R}, S_1, \ldots, S_m}[\mathsf{A}'(S_1, \ldots, S_m; r) = h_r] \geq 1 - \beta^2.$$

Employing Markov's inequality, we have

$$\Pr_{r \sim \mathcal{R}}\left[\Pr_{S_1, \ldots, S_m}[\mathsf{A}'(S_1, \ldots, S_m; r) = h_r] \geq 1 - \beta\right] \geq 1 - \beta.$$

In other words, the algorithm is $(1 - \beta, 1 - \beta)$-pseudo-globally stable as desired. Furthermore, observe that the answer of the distributional oracle is always within $\nu$ of the true answer; as a result, the accuracy guarantee of A also implies the accuracy of A$'$.

We now turn our attention to proving (6). Conditioned on $\mathscr{E}_{i-1}, \ldots, \mathscr{E}_1$, A issues the same $i$th query $\phi_i$ to both the empirical and the distributional versions of the oracle. From standard concentration inequality, with probability $1 - \frac{\beta^2}{3q}$, we have $|u_i - \hat{u}_i| \leq \frac{\beta^2}{3qI}$; the latter implies $d_{\mathrm{tv}}(\mathcal{P}_i, \hat{\mathcal{P}}_i) \leq \frac{\beta^2}{3q}$. Now, using the correlated sampling guarantee (Theorem 8) when this occurs, we have $\Pr_{r'_i}[\mathsf{CS}(\mathcal{P}_i; r'_i) \neq \mathsf{CS}(\hat{\mathcal{P}}_i; r'_i)] \leq \frac{2\beta^2}{3q}$. Applying a union bound then implies (6). $\qquad\square$

## B.2 Implications

While Theorem 5 may be applied to any of the many known SQ algorithms, we highlight two applications: *decision lists* (cf. [Riv87][6] and for definition) and *linear separators over* $\{0, 1\}^d$. Both classes are known to have efficient SQ algorithms [Kea98, DV08]. As a result, we obtain:

**Corollary 33.** *There exist $\varepsilon$-DP $(\alpha, \beta)$-accurate learners for decision lists and linear separators in the (public randomness) non-interactive local model of DP with $O\left(\frac{\mathrm{poly}(d) + \log(1/\beta)}{\varepsilon^2}\right)$ users, where each user has $\mathrm{poly}\left(\frac{d}{\beta\alpha}\right)$ samples. Moreover, these learners run in time $\mathrm{poly}(d, 1/\alpha, 1/\beta)$.*

This result is particularly interesting because our algorithm is *non-interactive* meaning that the users all just send the messages to the analyzer in one round. On the other hand, Daniely and Feldman [DF19] recently showed that in the item-level setting any non-interactive local DP learner requires exponential number of samples. This demonstrates the power of user-level DP learning in overcoming the non-interactivity barrier.

# C Lower Bounds

In this section we prove a lower bound (Lemma 4) on the number of users required in user-level private learning. The proof is essentially identical to that of [BNS19a] for the item-level setting.

We will need the following well-known bound often referred to as "group privacy":

**Lemma 34** (Group Privacy [DR14]). *Let A be any $(\varepsilon, \delta)$-DP algorithm and $O$ be any subset of outputs. Suppose that $D, D'$ are two datasets such that we can transform one to another by a sequence of at most $k$ addition/removal of users. Then, we have*

$$\Pr[\mathsf{A}(D) \in S] \leq e^{k\varepsilon} \cdot \Pr[\mathsf{A}(D') \in S] + \frac{e^{k\varepsilon} - 1}{e^\varepsilon - 1} \cdot \delta.$$

*Proof of Lemma 4.* We will prove the contrapositive. Suppose that there exists an $(\varepsilon, \delta)$-DP $(\alpha, 1/2)$-accurate learner A for the class $C$ that requires only $n \leq 0.01 \log(1/\delta)/\varepsilon$ users and each user has $m$ examples. We can construct a probabilistic representation $\mathcal{H}$ for $C$ as follows (where $\mathcal{H}$ denotes the probability of the output $H$):

---

**Algorithm 6** Constructing a Probabilistic Representation.

---

$H \leftarrow \emptyset$
**for** $T := 100\lceil \exp(\varepsilon n) \rceil$ times **do**
    Run A on the empty set (with no users) and add its output to $H$
Output $H$

---

From the construction, the size of $\mathcal{H}$ is at most $\log T = O(\varepsilon n)$ as desired.

---

[6]Note that what is commonly called decision lists (DL) today is called 1-DL by Rivest [Riv87], who also studied the generalization $k$-DL where each term in the list can be a $k$-conjunction. The results listed here also apply to $k$-DL but $d$ will be replaced by $d^k$.

To see that, $\mathcal{H}$ is an $(\alpha, 1/4)$-probabilistic representation of $\mathcal{C}$, consider any distribution $\mathcal{D}$; let $G := \{f \mid \mathrm{err}_{\mathcal{D}}(f) \leq \alpha\}$. From the guarantee of the learner, there must be sample sets $S_1, \ldots, S_m$ such that $\Pr[\mathsf{A}(S_1, \ldots, S_m) \in G] \geq 1/2$. Applying Lemma 34, we have

$$\Pr[\mathsf{A}(\emptyset) \in G] \geq \frac{1}{e^{n\varepsilon}} \cdot \left( \frac{1}{2} - \frac{e^{n\varepsilon} - 1}{e^{\varepsilon} - 1} \cdot \delta \right) \geq \frac{10}{T}.$$

Since we are running the learner $T$ times, the probability that at least one of them belongs to $H$ is at least 0.99. Thus, $\mathcal{H}$ is an $(\alpha, 0.01)$-probabilistic representation of $C$. $\qquad\square$

## D  Extension to Shuffle DP

In this section, we extend Theorem 1 to the shuffle model of DP, as stated below.

**Theorem 35.** *Let $\alpha, \beta \in (0, 0.1)$, and $C$ be any concept class with finite $\mathrm{LDim}(C) = d$. Then, for any $\varepsilon, \delta \in (0, 1)$, there exists an $(\varepsilon, \delta)$-shuffle-DP $(\alpha, \beta)$-accurate learner for $C$ that requires $O\left(\log(1/(\beta\delta))/\varepsilon\right)$ users where each user has $\tilde{O}_\beta\left((d/\alpha)^{O(1)}\right)$ samples.*

Recall that in the shuffle DP model [BEM+17, EFM+19, CSU+19], each user can produce a set of messages. The messages from all users are then randomly permuted together before being sent to the analyzer. Our goal is only to ensure that the shuffled messages satisfy $(\varepsilon, \delta)$-DP; when this holds, we say that the algorithm is $(\varepsilon, \delta)$-shuffle-DP. Similar to our result in the local model, here we assume that the users have access to shared (but not necessarily secret) randomness.

The only ingredient in the proof of Theorem 35 that is specific to the central model is the DP selection algorithm (Lemma 11). Hence, it suffices to prove a shuffle-DP selection algorithm with a similar guarantee, stated below.

**Lemma 36.** *For any $\varepsilon, \delta \in (0, 1)$ and $\beta \in (0, 0.1)$, there is an $(\varepsilon, \delta)$-shuffle-DP $(O(\log(1/(\beta\delta))/\varepsilon), \beta)$-accurate algorithm for the selection problem.*

We note that the above guarantee, unlike that of Lemma 11, does not have $\beta = 0$. However, this is just another "bad" event that can be included in the union bound.

In fact, our algorithm also works with for the *histogram* problem, where the setting is the same as selection except that we would like to output $\tilde{c}_u$ for every $u \in U$ and the error is defined as $\max_{u \in U} |\tilde{c}_u - c_u|$. Again, we say that an algorithm histogram is $(\alpha, \beta)$-accurate if the error is at most $\alpha$ with probability at least $1 - \beta$. Here we get:

**Lemma 37.** *For any $\varepsilon, \delta \in (0, 1)$ and $\beta \in (0, 0.1)$, there is an $(\varepsilon, \delta)$-shuffle-DP $(O(\log(n/(\beta\delta))/\varepsilon), \beta)$-accurate algorithm for the histogram problem.*

Selection, histogram, and related problems are well studied in the shuffle DP literature (e.g. [CSU+19, GMPV20, BBGN20, GKMP20, GGK+20, BC20, GGK+21, EFM+20, CZ21, GKM+21]). Currently, the best known algorithm for both problems yields error guarantees of either $O(\log |U|/\varepsilon)$ [GMPV20, BBGN20] or $O(\log(1/\delta)/\varepsilon^2)$ [BC20] for constant $\beta > 0$. Our Lemma 36 improves the latter to $O(\log(1/\delta)/\varepsilon)$ for selection and $O(\log(n/\delta)/\varepsilon)$ for histogram, both of which match the best known guarantees in the central model [KKMN09, BNS19b] when $\delta < 1/\mathrm{poly}(n)$.

### D.1  Selection Algorithm from Negative Binomial Noise

The remainder of this section is devoted to the proof of Lemma 36 and Lemma 37.

#### D.1.1  Binary Summation

We start by considering an easier problem of *binary summation* where each user $i$ receives an input $x_i$ and the goal is output an estimate $\tilde{s}$ of $\sum_{i \in [n]} x_i$. For this problem, we will prove the following:

**Lemma 38.** *For any $\varepsilon \in (0, 1), \delta, \beta \in (0, 0.5)$, there is an $(\varepsilon, \delta)$-shuffle-DP for the binary summation problem such that*

- *(Underestimation) The estimate $\tilde{s}$ is always at most the true value $\sum_{i \in [n]} x_i$.*
- *(Error Tail Bound) $\Pr\left[ |\sum_{i \in [n]} x_i - \tilde{s}| > O\left( \log\left( \frac{1}{\delta\beta} \right) / \varepsilon \right) \right] \leq \beta$.*

To prove Lemma 38, we recall the algorithm of [GKMP21], which adds noise drawn from the negative binomial distribution (denoted by $\mathrm{NB}(r, p)$) to the input; the randomizer and the analyzer[7] are presented in Algorithm 7 and Algorithm 8, respectively. Their privacy guarantee, proved in [GKMP21], is as follows[8]:

**Theorem 39** ([GKMP21, Theorem 13]). *For any $\varepsilon, \delta \in (0, 1)$, let $p = e^{-0.2\varepsilon}$ and $r \geq 3(1 + \log(1/\delta))$. Then, Algorithm 7 is $(\varepsilon, \delta)$-shuffle-DP.*

---

**Algorithm 7** Negative Binomial Randomizer (User $i$).

1: Samples $Z \sim \mathrm{NB}(r/n, p)$
2: Send $x_i + Z$ messages, where each message is 1

---

**Algorithm 8** Negative Binomial Analyzer.

1: **return** the total number of messages received

---

This almost immediately implies the following corollary.

**Corollary 40.** *For any $\varepsilon, \delta \in (0, 1)$, there is an $(\varepsilon, \delta)$-shuffle-DP for the binary summation problem such that*

- *(Overestimation) The estimate $\tilde{s}$ is always at least the true value $\sum_{i \in [n]} x_i$.*
- *(Error Tail Bound) $\Pr\left[|\sum_{i \in [n]} x_i - \tilde{s}| > O\left(\log\left(\frac{1}{\delta\beta}\right)/\varepsilon\right)\right] \leq \beta$.*

*Proof.* We simply use the Negative Binomial algorithm (Algorithms 7 and 8) with $p = e^{-0.2\varepsilon}$ and $r = \lceil 1000(1 + \log(1/(\beta\delta))) \rceil$. The privacy guarantee follows from Theorem 39. Next, we argue the utility guarantees.

- Notice that user $i$ sends at least $x_i$ messages. Thus, the total number of messages received by the analyzer is at least $\sum_{i \in [n]} x_i$.
- Since the summation of $n$ i.i.d. random variables drawn from $\mathrm{NB}(r/n, p)$ is distributed as $\mathrm{NB}(r, p)$, the total number of messages is $\sum_{i \in [n]} x_i + Y$ where $Y \sim \mathrm{NB}(r, p)$. Hence, to prove the error tail bound, it suffices to show that $\Pr_{Y \sim \mathrm{NB}(r,p)}\left[Y > O\left(\log\left(\frac{1}{\delta\beta}\right)/\varepsilon\right)\right] \leq \beta$. To see that this is true, we follow the approach in [Bro]. First, observe from the definition of the negative binomial distribution that, for any $T \in \mathbb{N}$, we have

$$\Pr_{Y \sim \mathrm{NB}(r,p)}[Y > T] = \Pr_{V \sim \mathrm{Bin}(T, 1-p)}[V < r],$$

  where $\mathrm{Bin}(\cdot, \cdot)$ denotes the binomial distribution. Now, we may select $T = 2r/(1 - p) = O\left(\log\left(\frac{1}{\delta\beta}\right)/\varepsilon\right)$. By applying the Chernoff bound, we have

$$\Pr_{V \sim \mathrm{Bin}(T, 1-p)}[V < r] \leq \exp(-0.001r) \leq \beta,$$

  which, as discussed above, implies the desired error tail bound. □

Note that Corollary 40 is not yet the same as Lemma 38 because the first guarantee is that the estimate is *at least* in the former instead of *at most* in the latter. However, it is simple to go from one to another, as formalized below.

*Proof of Lemma 38.* Run the algorithm from Corollary 40 but on input $1 - x_i$ (instead of $x_i$). Let $\tilde{s}$ denote the estimate of $\sum_{i \in [n]}(1 - x_i)$ from the algorithm. We then output $n - \tilde{s}$. The accuracy and privacy guarantees follow in a straightforward manner from that of Corollary 40. □

---

[7]We use a slightly different analyzer than the one in [GKMP21], where the mean of the noise was subtracted from the sum. We do not apply this step because we would like a lower bound of the true sum (Corollary 40).

[8]Note that [GKMP21, Theorem 13] only states the privacy guarantee for $r = 3(1 + \log(1/\delta))$. However, since $X + Y$ where $X \sim \mathrm{NB}(r^2, p), Y \sim \mathrm{NB}(r^1 - r^2, p)$ is distributed as $\mathrm{NB}(r^1, p)$ for $r^1 > r^2 > 0$, the algorithm for larger $r$ is only more private since it can be thought of as post-processing of that of a smaller $r$.

### D.1.2   From Binary Summation to Selection

Now that we have proved Lemma 38, let us note that it easily implies Lemma 36 by running the binary summation protocol "bucket-by-bucket" as described below.

*Proof of Lemma 36.* The selection algorithm works by running the $(\varepsilon/2, \delta/2)$-shuffle-DP binary summation algorithm for each bucket in parallel[9] where in bucket $u \in U$, we let $x_i^u = \mathbf{1}[u_i = u]$. Let $\tilde{c}_u$ denote the output estimate of $\sum_{i \in [n]} x_u^i = c_u$ of bucket $u$. We finally output $u^* = \arg\max_{u \in U} \tilde{c}_u$.

The privacy guarantee of the algorithm follows from the fact that changing a single $u_i$ effects at most two buckets; thus, the basic composition implies that the above algorithm is $(\varepsilon, \delta)$-shuffle-DP.

We now argue its accuracy guarantee. Let $u^{\mathrm{opt}} = \arg\max_{u \in U} c_u$. From the second guarantee of Lemma 38, we have $\tilde{c}_{u^{\mathrm{opt}}} \geq c_{u^{\mathrm{opt}}} - O\left(\log\left(\frac{1}{\delta\beta}\right)/\varepsilon\right)$ with probability $1 - \beta$. When this holds, the first property of Lemma 38 ensures that we output $u^*$ such that

$$c_{u^*} \geq \tilde{c}_{u^*} \geq \tilde{c}_{u^{\mathrm{opt}}} \geq c_{u^{\mathrm{opt}}} - O\left(\log\left(\frac{1}{\delta\beta}\right)/\varepsilon\right).$$

This means that the algorithm is $\left(O\left(\log\left(\frac{1}{\delta\beta}\right)/\varepsilon\right), \beta\right)$-accurate as desired. $\qquad\square$

### D.1.3   From Binary Summation to Histogram

The histogram protocol is similar to above except that we use the failure probability $\beta/n$ and we output zero instead of negative estimates. (This is similar to the protocols in the central model.)

*Proof of Lemma 37.* Similar to the proof of Lemma 37, the histogram algorithm runs the $(\varepsilon/2, \delta/2)$-shuffle-DP binary summation algorithm with failure probability $\beta/n$ for each bucket in parallel where in bucket $u \in U$, we let $x_i^u = \mathbf{1}[u_i = u]$. Let $\tilde{c}_u$ denote the output estimate of $\sum_{i \in [n]} x_u^i = c_u$ of bucket $u$. Finally, for any bucket such that $\tilde{c}_u < 0$, we let $\tilde{c}_u = 0$ instead.

The privacy guarantee holds due to the same reason as in the proof of Lemma 37.

We now argue its accuracy guarantee. Let us divide the buckets into two types: $U_{>0} := \{u \in U \mid c_u > 0\}$ and $U_{=0} := \{u \in U \mid c_u = 0\}$.

- For any $u \in U_{=0}$, due to the first property of Lemma 38, we always output zero and thus the error here is zero.
- Now consider any $u \in U_{>0}$. From our choice of parameters and the second property of Lemma 38, we have $\Pr\left[|c_u - \tilde{c}_u| > O\left(\log\left(\frac{n}{\delta\beta}\right)/\varepsilon\right)\right] \leq \beta/n$.

Note also that $|U_{>0}| \leq n$. As a result, we may apply a union bound over all $u \in U_{>0}$ and conclude that the error is at most $O\left(\log\left(\frac{n}{\delta\beta}\right)/\varepsilon\right)$ with probability $1 - \beta$. $\qquad\square$

---

[9]Note that, since all messages are shuffled together, we have to append $u$ to the beginning of the messages to indicate that the message corresponds to bucket $u$; see e.g., [GKMP20, Appendix B] for a more detailed description.