# OpenReview forum: "User-Level Differentially Private Learning via Correlated Sampling"
_NeurIPS.cc/2021/Conference — NeurIPS 2021 Poster_

### Official Review · Reviewer_kC68 · 2021-07-17

**Rating:** 8
**Confidence:** 2

**Summary:**

This work tackles the problem of PAC learning with user-level privacy guarantees. The paper's primary contribution is a nearly tight characterisation of the number of users required when the learning is via an 1) $\epsilon$-DP and 2) $(\epsilon,\delta)$-DP algorithm. At the core of the paper is the idea that users can share randomness before the sampling stage. This insight allows for the learnability of all online learnable classes, with much fewer users(exponentially fewer in terms of $d$).

**Limitations And Societal Impact:**

Yes, the authors have adequately discussed the limitations and the societal impact.

**Main Review:**

The paper is quite well written. The work presents a creative and original connection between correlated sampling and DP learning. The implications of the assumption are explored in-depth.

**Time Spent Reviewing:**

4

---

> ### Author Response · Authors · 2021-08-09
> **Response**
>
> Thank you for your review.

---

### Official Review · Reviewer_xkiJ · 2021-07-17

**Rating:** 7
**Confidence:** 3

**Summary:**

This paper studies user-level privacy for any privately PAC learnable class in the setting where each user holds more than one training example. It first shows that for any online learnable class (with finite Littlestone dimension), it's possible to learn the class with an (\eps,\delta)-DP algorithm using only O(\log(1/\delta)/\eps) users. Then it gives sample complexity of a generic \eps-DP learner in both the local and the central models. Finally, it shows that we can take any efficient SQ algorithm and turn it into an efficient learner in the user-level setting. The protocols are based on a novel connection between correlated sampling and DP learning.

**Limitations And Societal Impact:**

The paper discussed the limitations and proposed several open questions.

**Main Review:**

This is a well-written paper. I enjoy reading it. Protecting user-level privacy has raised much attention recently, and it's timely research. I particularly like the idea of exploiting the shared randomness between users, which is handled by using correlated sampling. It's nontrivial to connect the correlated sampling technique with DP learning. The theoretical results and algorithms are sound. My only question is in Algorithm 1, how do you select A? Can it be any learners satisfying that A's output consists only of hypotheses with (\alpha, \beta/k)-accuracy?

Minor comment: A type in line 35 ``artifically''

**Time Spent Reviewing:**

6

---

> ### Author Response · Authors · 2021-08-09
> **Response**
>
> Thanks for your review.  We will fix the typos.
>
> **Selecting A.**
> According to the assumptions of Theorem 20, $\mathsf{A}$ can be any learner that is accurate and list-globally stable.

---

### Official Review · Reviewer_4gym · 2021-07-18

**Rating:** 7
**Confidence:** 4

**Summary:**

This paper studies user level differential privacy in the setting of PAC learning. They study both central and local model of differential privacy and both approximate and exact differential privacy. In the central model, they show that they only need log(1/\delta)\epsilon users and Ldim^O(1) samples per users to guarantee (\epsilon, \delta)  differential privacy whereas in the item level DP, one needs Ldim^O(1) samples.  They also provide nearly matching lower bounds on the number of users.

**Limitations And Societal Impact:**

Yes

**Main Review:**

I think the paper studies an interesting problem of DP guarantees for user level setting in the PAC model. Generally, there is a lot of interest in differentially privacy and it would be of interest to the community. This paper obtains the first bounds in the user level model in PAC setting. It shows an interesting result that we only require users  depending on epsilon, delta for (epsilon, delta) DP independent of the Littlestone dimension and the number of samples depend on the Littlestone dimension. This paper builds on existing connections between global stability and differential privacy and additionally, uses the technique of correlated sampling. This idea of using correlated sampling in this setting seems interesting to me. The paper is mostly well written.

Some questions -

1) I don’t understand the statement in line 35 “One way to understand this problem is to artificially limit the number of training examples contributed by each user.” when the later statement says that DP cost is less when each user has large number of samples.
2) Is this the first paper studying user level DP in the PAC learning setting?
3) The authors mention that there is a gap between user level and item level privacy in the local model. What is the relationship between SQ-dim and PRdim? The authors mention that SQ-dim can be bigger. Is it always lower bounded by PRdim?
4) Is there any general connection or relationship that one can obtain between the user level and item level bounds?

**Time Spent Reviewing:**

3

---

> ### Author Response · Authors · 2021-08-09
> **Response**
>
> **Line 35.**
> We apologize for the confusing wording.  What we meant to say is that limiting the contribution from each user is an easy way to translate results for item-level privacy to user-level privacy; this approach views multiple examples per user as an overhead.  Our work shows the converse: that it is possible to exploit the fact that each user has many examples.
>
> **PAC setting.**
> To the best of our knowledge, this is the first work studying user-level DP in the PAC setting.
>
> **SQDim vs PRDim.**
> We are *not* aware of a formal relationship between SQDim and PRDim. However, Feldman and Xiao (COLT 2014) showed that PRDim is equivalent to the one-way communication complexity of the concept class. It *might* be possible to use this to prove that SQDim is lower-bounded by PRDim since an algorithm that uses few SQ queries should (intuitively) require low communication. We will investigate this further and mention this as an interesting open question in case we do not manage to prove the formal relationship.
>
> **General connection.**
> For central DP, the total number of items needed in the user level setting is at least that of the item level setting; this can be seen because any user-level DP algorithm is also item-level DP.
>
> Despite the above connection, the situation in local DP is much more interesting because a user-level DP algorithm does not correspond to any algorithm for the item-level setting. Indeed, there does not seem to be a formal relationship in this setting, because we manage to give an algorithm whose sample complexity is substantially smaller in the former case (e.g., for PARITY), which, as explained in the previous paragraph, is not possible in the central DP setting.

---

### Official Review · Reviewer_R7Bc · 2021-07-19

**Rating:** 7
**Confidence:** 2

**Summary:**

The setting in the paper is PAC learning with user-level DP, where the main difference from previous works is that each user may have many examples (instead of only a single example).
The main result is that it is possible to learn [(epsilon,delta)-DP] with a small number of users as long as each user has sufficiently many samples.
Another main result is epsilon-DP learner in the local model.
Moreover, the authors provide nearly matching lower bounds.

**Limitations And Societal Impact:**

-

**Main Review:**

Significance: It seems like the paper discusses some fundamental questions in Learning with DP guarantees. The results are significant and interesting.
It seems like the techniques introduced in the paper might be of independent interest.
It is hard for me to fully understand the motivation of this setting, due to some lack of knowledge of previous results.

Clarity and relation to prior work: The paper is very clear and easy to follow. Also, the literature review is thorough.

It was quite hard for me to check some proofs, due to lack of previous technical background.
I tend to accept the paper, but I will keep my confidence low.

**Time Spent Reviewing:**

3

---

> ### Author Response · Authors · 2021-08-09
> **Motivation**
>
> **Motivation.**
>
> As we indicated in the paper, the main motivation for studying this setting comes when there are multiple training examples contributed by each user.  This is common in settings such as federated learning.  For example, consider the problem of building a private model for web search click prediction, where the training examples are based on user clicks on search results.  It is easy to see that in this setting, each user can contribute to multiple training examples.  Another example is building models for suggesting queries (eg, the Gboard model from Google); the training examples are based on historical queries issued by the user.  Once again, each user can contribute multiple training examples and we would like to maintain privacy at the level of the user.  In fact, studying the tradeoff between item- and user-level privacy, for private learnability, is one of the important questions raised in the well-known survey of Kairouz et al [KMA+19] on Federated Learning.

---

### Official Review · Reviewer_R9c6 · 2021-07-30

**Rating:** 7
**Confidence:** 3

**Summary:**

Given a dataset consisting of $n$ users, where each user $i$ has a sequence $(x^i_1, y^i_1), \ldots, (x^i_m, y^i_m)$ of $m$ samples, the goal of this paper is to investigate differentially private learning in three settings: central-pure ($\epsilon$-DP), central-approximate (($\epsilon, \delta$)-DP), and local-pure.

Using results on correlated sampling, global stability, and private selection algorithms, the authors built efficient algorithms for the three above-mentioned settings. The key idea is to have each user first "learn" insights from their $m$ samples (and obtain a local hypothesis candidate), and then use DP selection algorithm (e.g., the Exponential Mechanism) to find the best hypothesis. Finally, the authors give lower bounds for the three DP models.

**Limitations And Societal Impact:**

N/A: In Checklist, the authors said they describe the limitations in Section 5; but Section 5 only includes conclusions and several open questions for future work. The authors did not mark any potential negative societal impacts.


**Main Review:**

Overall, I think the authors made a great contribution towards the problem of differentially private learning where each user has $m$ samples. I like the comprehensive results and the systematic investigation of the problem.


Some (minor) issues:
- Practicality: while from a theoretical point of view, the paper made significant contribution, it is unclear to me how practical the algorithms are. For example, how to learn a hypothesis from samples, how to encode hypotheses into the DP selection algorithms. It would be great if the authors can explain them more clearly and even provide corresponding code.
- The relationship between item-level and user-level: in this paper, item-level DP means each user has only one item. But more commonly, item-level DP assumes each user has some number of items, but the definition of neighborhood Is different (only one item is different). I understand that the authors might want to avoid confusion, but this could still confuse those who are familiar with DP. A better way of presentation could be changing item-level and user-level to single-value setting and multi-value setting, and clarifying the difference between item-level and user-level protection (e.g., in the graph setting, item-level definition protects the existence of an edge, while user-level definition protects the existence of a node).
- Theorems 2 and 3 both mention the local model: Theorem 3 should be in the central model?
- SQ is not spelled out.


Disclaimer: I am outside of the NeurIPS community and haven't read a lot NeurIPS papers. This paper is a clear acceptance for me, but I cannot rank it among other papers.



**Time Spent Reviewing:**

5

---

> ### Author Response · Authors · 2021-08-09
> **Response**
>
> **Practicality.**
> Our primary emphasis in this work is theoretical.  Our $(\epsilon, \delta)$-DP algorithm (Theorem 1) adds very little overhead (see Algorithm 1) on top of the underlying list-stable learner $\mathsf{A}$, since Correlated Sampling by itself is quite practical (eg, it is used in data summarization/sketching applications).  So, as long as $\mathsf{A}$ is practical, our algorithm will remain practical.  We also remark that any underlying learner we use has to encode the hypothesis (when it outputs) anyway, so we may use the same encoding as the underlying learner.
>
> Our $\epsilon$-DP algorithms (Theorems 2, 3) are not efficient (see the discussions at the end of page 2 and start of page 3), which is part of the reason we prove Theorem 5, which is efficient as long as the underlying SQ learner is efficient. We think it is an interesting question on how to get additional $\epsilon$-DP efficient learners and we discuss this in the second paragraph of the Conclusions section.
>
> **Single-value and multi-value.**
> Thanks for the suggestion.  We will consider incorporating it.
>
> **Theorem 2, 3, SQ.**
> Theorem 2 is in the local model and Theorem 3 is in the central model (please see Table 1).   (Current Theorem 3 statement has a typo and it should refer to the central model.)
> We will make these explicit in the theorem statements in the revision.  We will also move SQ definition (currently Definition 31 in the Supplementary Material) to the main body.
>
> **Limitations.**
> We apologize for not clearly marking the limitations in Section 5.  As stated there, one of the limitations of our current work is the use of shared randomness.  We will make this explicit in our discussion.

---

> > ### Comment · Reviewer_R9c6 · 2021-09-10
> > **Thank you for the response**
> >
> > Thank you for the response. My comments are addressed.

---

### Decision · Program_Chairs · 2021-09-28

**Decision:**

Accept (Poster)

**Comment:**

All the reviewers greatly appreciate the significance and novelty of this work, with some minor issues raised. For example,
- making the statement of some theorems more clear,
- motivating this work better,
- explaining the practicality,

and so on. Please try the best to address these issues in the final version

**Consistency Experiment:**

NeurIPS has a long history of experimentation. In 2014, NeurIPS ran an experiment in which 10% of submissions were reviewed by two independent committees to quantify the randomness in the review process. This year, we repeated a variant of this experiment to see how the quality of the review process has changed over time.  This paper was part of the experiment and was therefore assigned to two committees (consisting of reviewers, an Area Chair, and a Senior Area Chair) that reached independent decisions.  If both committees made the same recommendation, this recommendation was followed. If a single committee recommended acceptance, the paper was accepted (with the exception of a few cases in which the other committee identified what we considered a fatal flaw, e.g., an error in a key result).

Both committees reached the same decision: **Accept (Poster)**

The other committee assigned to the paper recommended **Accept (Poster)**.  You can find the other set of reviews, along with any follow up discussion with the authors here:
https://openreview.net/forum?id=PqiCvohYSAx